# Multiple scattering dynamics of fermions at an isolated p-wave resonance

R. Thomas[1], K.O. Roberts[1], E. Tiesinga[2], A.C.J. Wade[1,3], P.B. Blakie[1], A.B. Deb[1] & N. Kjærgaard[1]

The wavefunction for indistinguishable fermions is anti-symmetric under particle exchange, which directly leads to the Pauli exclusion principle, and hence underlies the structure of atoms and the properties of almost all materials. In the dynamics of collisions between two indistinguishable fermions, this requirement strictly prohibits scattering into 90° angles. Here we experimentally investigate the collisions of ultracold clouds fermionic $^{40}K$ atoms by directly measuring scattering distributions. With increasing collision energy we identify the Wigner threshold for p-wave scattering with its tell-tale dumb-bell shape and no 90° yield. Above this threshold, effects of multiple scattering become manifest as deviations from the underlying binary p-wave shape, adding particles either isotropically or axially. A shape resonance for $^{40}K$ facilitates the separate observation of these two processes. The isotropically enhanced multiple scattering mode is a generic p-wave threshold phenomenon, whereas the axially enhanced mode should occur in any colliding particle system with an elastic scattering resonance.

[1] Department of Physics, QSO—Centre for Quantum Science, and Dodd-Walls Centre for Photonic and Quantum Technologies, University of Otago, 730 Cumberland Street, Dunedin 9016, New Zealand. [2] Joint Quantum Institute and Center for Quantum Information and Computer Science, National Institute of Standards and Technology and University of Maryland, Gaithersburg, Maryland 20899, USA. [3] Department of Physics and Astronomy, Aarhus University, DK-8000 Aarhus C, Denmark. Correspondence and requests for materials should be addressed to N.K. (email: niels.kjaergaard@otago.ac.nz).

The Pauli exclusion principle[1], which forbids two indistinguishable fermions from occupying the same quantum state, is one of the most ubiquitous concepts in physics, underpinning the periodic table, electron transport in condensed-matter systems and the astrophysical phenomena of white dwarf and neutron stars. It arises from the unique quantum phenomenon of indistinguishable particles and a primary difference between the two fundamental types of particles—bosons and fermions—lies in their respective symmetrization requirements. Given a scattering amplitude $f(\theta)$ with polar angle $\theta$ relative to the collision axis, the properly symmetrized differential cross-section for indistinguishable particles is

$$\frac{\mathrm{d}\sigma}{\mathrm{d}\Omega} = |f(\theta) \pm f(\pi - \theta)|^2, \qquad (1)$$

where the symmetric version is for bosons and the anti-symmetric one is for fermions[2]. As a result, there can be no scattering of indistinguishable fermions into angles 90° from the axis[3], regardless of collision energy or the details of the scattering potential. For rotationally invariant interactions, one can perform a partial-wave expansion of $f(\theta)$ in a sum over Legendre polynomials $P_\ell(\cos\theta)$ indexed by the orbital angular momentum $\ell$. As the $\ell=0$ partial wave is symmetric, the lowest order contribution to the differential cross-section comes from the $\ell=1$ partial wave $f(\theta) \propto \cos\theta$, giving rise to scattering halos[4–9] akin to the 'dumb-bell'-shaped p-wave orbitals in atoms. For sufficiently short-ranged potentials, such as those between non-magnetic atoms, p-wave scattering is strongly suppressed[10].

Single event dynamics are the starting point when considering the general scattering problem; however, in many situations the response of the particles to the scattering potential has significant contributions from multiple events. Multiple scattering is known to lead to intriguing effects such as Anderson localization[11–13], coherent backscattering[14,15] and random lasing[16] in the domain where atoms have a long de Broglie wavelength and a wave-like description. One may ask whether similar, non-trivial, dynamics can arise in a semiclassical regime, where atoms behave similar to particles. In this regard, indistinguishable fermions offer a pristine environment in which to investigate multiple scattering, because it is possible to restrict collisions to a single partial wave—the anisotropic p-wave—over a wide range of energies. This implies that the inherent binary scattering anisotropy is energy independent, and that energy-dependent deviations in the angular pattern of the scattering halo from its fundamental p-wave shape become a sensitive measure of multiple scattering. In particular, atoms emerging at 90° to the collision axis present a telltale signature for multiple scattering.

In this study, we employ an optical collider[17] to investigate multiple scattering of ultracold clouds of fermionic $^{40}$K in the neighbourhood of an isolated p-wave scattering resonance. We find that above a threshold collision energy the resonance establishes two distinct regions with multi-scattering dynamics driven by cascades to lower collision energies. These regions are identified and contrasted via their isotropic and anisotropic contributions to the underlying pure p-wave binary scattering pattern. Our ability to classify and interpret the scattering dynamics via these residuals is crucially facilitated by every scattering event being exclusively p-wave.

## Results

### Apparatus and experimental procedure.
Using a steerable crossed-beam optical dipole trap, we collide clouds of $^{40}$K atoms with temperatures of $T = 1.3 \pm 0.1\,\mu\text{K}$ at comparatively high energies $E/k$ from 50 to $1,800\,\mu\text{K}$ (measured in units of the Boltzmann constant $k$). A horizontal laser beam supports the atoms against gravity and provides some transverse confinement,

whereas a vertical beam is steered by an acousto-optic deflector[18] (see Methods for details on the experimental sequence). We initially split a single cloud of $\sim 10^6$ atoms in the $|F = 9/2, m_F = 9/2\rangle$ hyperfine state into two clouds separated by a controllable distance (Fig. 1a) and then accelerate those clouds towards each other (Fig. 1b). Shortly before the collision, the dipole trap is turned off so that the atoms collide in free space (Fig. 1c) and the atom clouds and scattering halo are allowed to expand for a variable amount of time before imaging (Fig. 1d). For expansion times much longer than the ratio of initial cloud size to cloud speed ($>100\,\mu\text{s}$ for our lowest energy), the position distribution is a scaled version of the momentum distribution, which would not be the case if collisions occurred in trap[19].

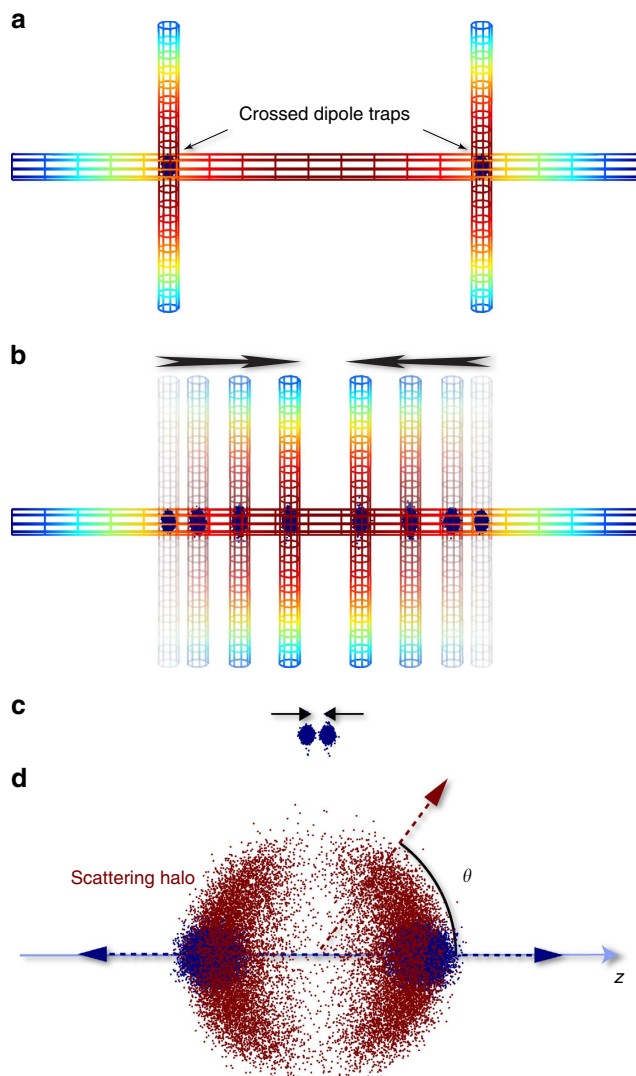

**Figure 1 | Collision sequence.** (**a**) A pair of ultracold atom clouds (blue dots representing atoms) are confined by two crossed optical dipole traps at the intersections of a horizontal and two vertical laser beams. The laser beams are shown as a coloured mesh with red, indicating a high field intensity. (**b**) By steering the two vertical laser beams the atomic clouds are accelerated towards each other. (**c**) Shortly before collision, both the vertical and horizontal light fields are switched off so that collisions occur in free space. (**d**) After the collision, the clouds expand and move away accompanied by a halo of scattered atoms (red dots) expanding into a p-wave halo centred on the collision point. $\theta$ denotes the polar angle with respect to the axis of collision ($z$).

**Two-body $^{40}$K scattering at low energies.** In Fig. 2a, we plot the scattering cross-section $\sigma(E)$ for two $^{40}$K atoms in the $|F = 9/2, m_F = 9/2\rangle$ state as a function of centre-of-mass collision energy, calculated using a coupled-channels model with parameters from ref. 20. Two features are of note. The first is the p-wave shape resonance[21,22] at $E_{res}/k \approx 350\,\mu$K. The second feature is the threshold behaviour $\sigma(E) \propto E^2$ as $E \to 0$, resulting from the Wigner threshold law for partial cross-sections[23]

$$\sigma_\ell(E) \underset{E \to 0}{\propto} E^{2\ell}, \qquad (2)$$

for $\ell = 1$ in a van der Waals potential. With even $\ell$ contributions to scattering strictly absent from anti-symmetrization (equation (1))—in particular that of $\ell = 0$ (the isotropic s-wave)—no atomic scattering can occur when the collision energy approaches zero[24,25]. As $\sigma_l$ for $\ell = 3, 5, \ldots$ falls off much faster than $\sigma_1$ as $E \to 0$ (equation (2)), $\ell = 1$ will dominate and an extended low-energy range exists where, essentially, pure p-wave scattering reigns.

**Observations.** Figure 2 includes experimental absorption images of scattering halos for five different energies along with their angular scattering distributions (for details on image processing, see Supplementary Note 1). At $E/k = 150\,\mu$K (Fig. 2b,g), we observe an angular distribution that is consistent with a pure p-wave scattering halo. In particular, we note the extinction of 90° scattering, characteristic of indistinguishable fermions. This behaviour extends up to energies $E/k \lesssim 180\,\mu$K (Fig. 2c,h), which we take to define the upper boundary of a suppression region where the threshold behaviour of the cross-section prevents higher-order scattering. For collision energies above this suppression region and in the vicinity of the shape resonance, we encounter a significant reduction in the contrast of the scattering distribution. The distributions are reminiscent of a p-wave halo riding on top of a constant offset, which corresponds to isotropic scattering. Figure 2d,i present the case of $E/k = 300\,\mu$K. As scattering of indistinguishable fermions enforces a restriction to odd-$\ell$ partial waves, the emergence of scattered atoms at 90° cannot result from primary scattering events.

Assuming that all scattering is p-wave and that higher-order events have a collision axis that is rotated from the original by a small, random angle $\theta_0$ with zero mean, the angular distributions of these events will have the form $\langle \cos^2(\theta - \theta_0) \rangle = (1 - 2\langle\theta_0^2\rangle)\cos^2\theta + \langle\theta_0^2\rangle$. This adds to the underlying p-wave angular distribution $\propto \cos^2\theta$, altogether giving an expression for angular scattering

$$D(\theta) = a\cos^2\theta + b, \qquad (3)$$

where $a$ and $b$ are constants, which depend on $\langle\theta_0^2\rangle$ and the number of primary and multiply scattered particles. When fitting equation (3) to our experimental data, we indeed obtain good agreement for energies up to $\sim 800\,\mu$K, which we take to be the upper bound of an isotropic enhancement region. The hallmark of this region is an isotropic scattering background, captured in the parameter $b$, on which the primary p-wave scattering resides. Curiously, for energies higher than $\sim 800\,\mu$K scattering at 90° vanishes, whereas at the same time a p-wave angular distribution fails to describe our data; the cases $E/k = 1,380\,\mu$K and $E/k = 1,660\,\mu$K are presented in Fig. 2e,j and Fig. 2f,k, respectively. The measured angular scattering has a wider trough near $\theta = \pm 90°$ and more scattering into $\theta = 0, 180°$ than contained in the simple $\cos^2\theta$ dependence of equation (3); we define this region as the axial enhancement region. The left–right asymmetry of the data in Fig. 2e,f,j,k results from atoms escaping the two traps unevenly during the acceleration phase. This effect is caused by the diffraction efficiency and mode structure of the acousto-optic

deflector, which generates the time-averaged optical traps, not being entirely symmetric about the collision point. It has previously been noted[19] that the collision between differently sized clouds may break inversion symmetry for the scattering halo via multiple scattering.

**Numerical simulations.** To gain insights into the mechanism leading to three distinct regions of scattering, we employ the

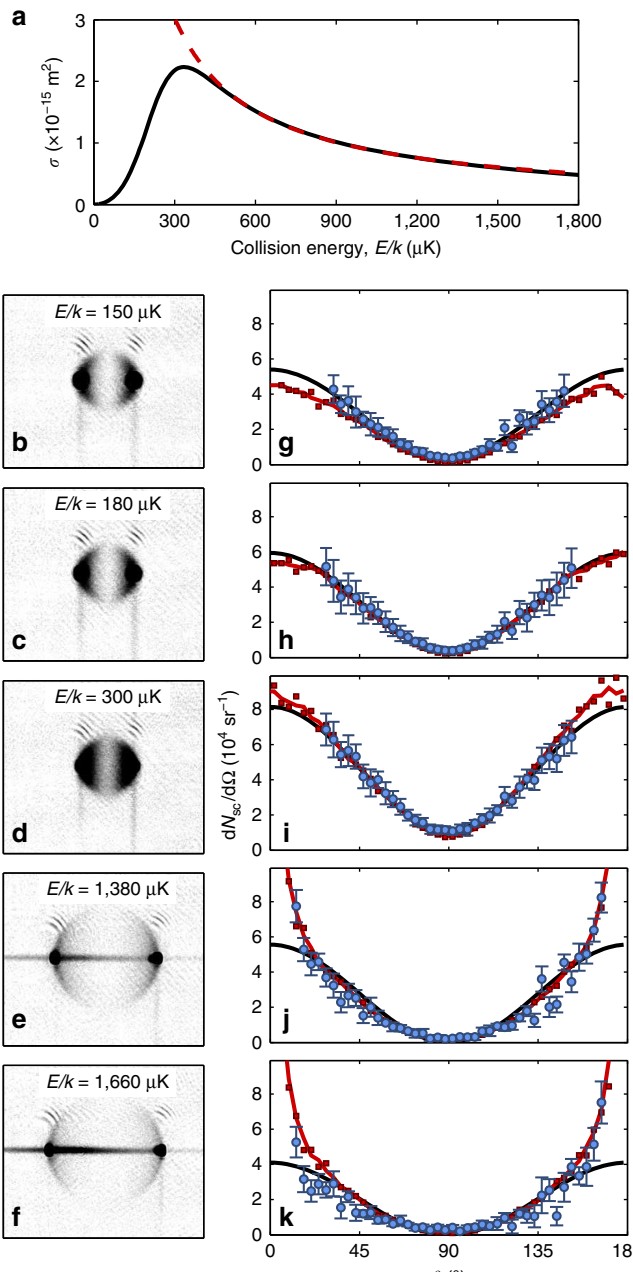

**Figure 2 | Scattering halos and analysis.** (a) Total cross-section for spin polarized $^{40}$K collisions as a function of energy (black solid line) with unitarity bound (red dashed line). (b–f) Grayscale absorption images of scattering halos at the specified collision energies. (g–k) Corresponding measured angular scattering yield $dN_{sc}/d\Omega$ as a function of angle $\theta$ to the collision axis shown as blue circles with error bars denoting the s.e. of the mean (statistical) plus a 4% systematic error. The scattering distributions are modelled by fitting the data to equation (3) (black line) and through numerically simulated scattering distributions (red squares and line) obtained by means of the DSMC method.

direct-simulation Monte–Carlo (DSMC) method for numerically integrating the Boltzmann equation describing our experiment[26] (see Methods for details). We convert the final positions of the atoms given by the DSMC simulation into synthetic absorption images, which we then analyse using the same methods as for the experimental images. Simulated angular scattering distributions from a model with only the total number of atoms as a global fit parameter (see Supplementary Note 2) are plotted in Fig. 2g–k and we obtain good agreement at all energies between the simulation and experiment. In particular, the simulation captures the nontrivial axial enhancement region.

To further verify that the DSMC method models the dynamics, we compare the number of atoms scattered by the collision $N_{sc}$ as a function of the energy in Fig. 3a as found in experiment and simulation, with the total number of atoms determined by the aforementioned global fit. The number of scattered atoms is calculated in the same way for both experimental and simulated data by integrating over fits to $D(\theta)$ (equation (3)), which allows us to interpolate over the dense, unscattered clouds. We obtain excellent agreement between our experimental data and the simulation with slight deviations at low and high energies; these arise from a combination of technical noise, imperfect state preparation and finite trap depth (see Supplementary Note 2). In addition to the total number of scattered atoms, we also display

the DSMC prediction for the number of atoms that experience one and only one scattering event, that is, those that would generate a pure p-wave halo. The difference between this curve and the total is due entirely to multiply scattered atoms. Only for $E_{seed} \lesssim 180\,\mu K$, in the suppression region, can the scattering halos be explained solely in terms of single scattering events.

**Classification of multiple scattering dynamics.** To understand the underlying dynamical process by which the scattering distributions separate into three different regions (I: Suppression, II: Isotropic enhancement and III: Axial enhancement; see Fig. 3), we plot in Fig. 3b the distribution of collision energies $E_{coll}$ generated by the collision of two clouds at a seed energy $E_{seed}$ as extracted from the DSMC simulation. In what follows, we detail the energy considerations in the simplified case of secondary collisions between one atom that has never scattered and one atom that has experienced only one scattering event. The collision energy for this type of collision is always bounded from above by $E_{seed}$, as the corresponding momentum shells have radii smaller than that of primary collision at the seed energy as illustrated geometrically in Fig. 4.

For region I, extending from the Wigner threshold and below, where the scattering cross section falls off with decreasing energy as $\sigma(E) \propto E^2$, secondary events at $E_{coll} < E_{seed}$ are suppressed with respect to primary collisions occurring at the seed energy. The full distribution of $E_{coll}$, shown in Fig. 3b (region I), is approximately symmetrical about $E_{seed}$ with an s.d. $\sqrt{2E_{seed}kT}$. The momentum space distribution is dominated by the primary collisions and is shown in Fig. 4a.

In contrast, collisional halos seeded at an energy within region II acquire an isotropic component. At a broad range of energies centred on the shape resonance, secondary scattering occurs only if the relative energy of the new collision pair is close to that of the original event. The velocities must be nearly anti-parallel—implying that the centre-of-mass velocity must be small—and there will be little difference between the scattering halo in the centre-of-mass frame and the halo in the laboratory frame. The most significant effect of the initial velocities not being perfectly anti-parallel is that the collision axis will be slightly rotated compared with the original direction, as shown in Fig. 4b. As noted previously, a small random rotation of the collision axis gives rise to, over many realizations, an isotropic halo. Collision energies will be lower than that of the original event, yielding energy distributions that are wider than predicted for single scattering events and skewed towards low energies as seen in region II of Fig. 3b.

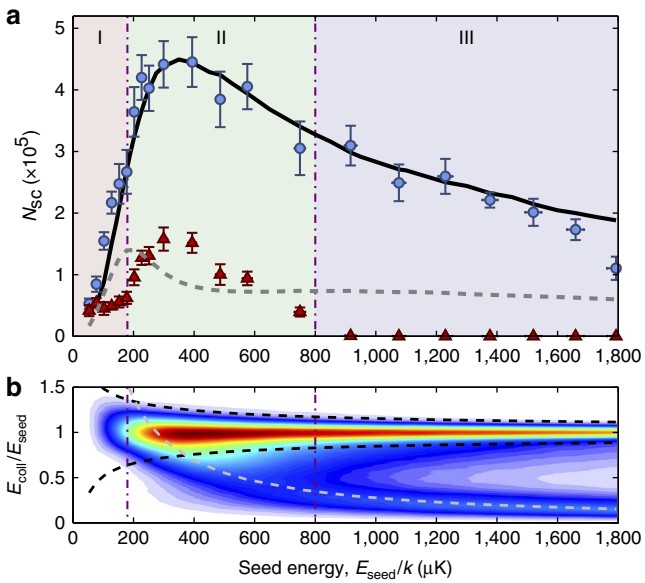

**Figure 3 | Number of scattered atoms as a function of seed energy.**
(**a**) Blue circles are the mean of five independent measurements of the number of scattered atoms $N_{sc}$ from five separate images and the error bars are the s.e. of the mean thereof (statistical) plus a 4% systematic error in determining the absolute number of atoms. Red triangles are the mean value of the isotropic scattering fraction from the fit of $D(\theta)$ (equation (3)) to the experimental data. The black solid line indicates the DSMC simulation of the experiment assuming a total atom number of $8.2 \times 10^5$. The dashed grey line is the prediction of the DSMC model for the number of atoms that have experienced only one scattering event. Purple dash-dotted lines delineate the three collision regions I: Suppression, II: Isotropic enhancement and III: Axial enhancement. (**b**) Density plot of the distribution in collision energies $E_{coll}$, normalized to seed collision energy $E_{seed}$, with $E_{coll}/E_{seed}$ on the ordinate and $E_{seed}$ on the abscissa. Colour denotes the number of collisions at a given point, with white colouring indicating zero collisions progressing to red as an arbitrary maximum. Black dashed lines indicate thrice the expected s.d. of the $E_{coll}/E_{seed}$ distribution when only single scattering events are present. The light grey dashed line is the contour $E_{coll}/k = 275\,\mu K$.

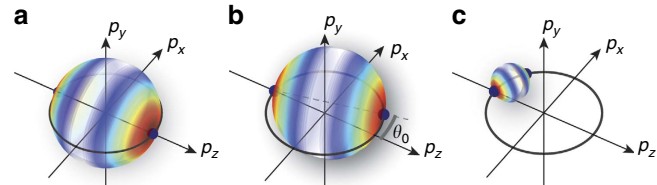

**Figure 4 | Momentum space representation of primary and secondary scattering.** (**a**) Primary collisions in the suppression region occur between atoms (solid dark blue spheres) that have equal and opposite momenta. Collisions redistribute momenta onto a spherical shell weighted by the single-particle differential cross-section; here, red is a high value, whereas white is zero. (**b**) Collisions between atoms that are nearly counter-propagating occur at lower collision energies (smaller momentum shell radius) and with a collision axis slightly rotated by an angle $\theta_0$. (**c**) Collisions between atoms that are nearly co-propagating occur at much lower collision energies and all final momenta lie close to the original collision axis.

Finally, region III describes a regime for axially enhanced multiple scattering. Here, seed energies are much higher than $E_{res}$ and it is possible for lower-energy secondary collisions to be resonantly enhanced. A bimodal collision energy distribution appears, as in region III of Fig. 3b, with the high-energy peak being centred on $E_{coll} \approx E_{seed}$ with a width determined primarily as for region I and the low-energy peak being centred on $E_{coll} \approx 275\,\mu K$. This energy is lower than $E_{res}$ due to competition between a higher cross-section at $E_{res}$ and more atoms originating at shallower angles, leading to lower collision energies. These low collision energies only occur when the velocities of the collision partners are nearly parallel and hence the centre-of-mass velocity is correspondingly large (Fig. 4c). In the laboratory frame, the scattered atoms travel nearly parallel to the original collision axis and this leads to an enhancement of atoms scattered into angles close to $\theta = 0$, 180° and a reduction in the number of atoms available to be scattered into angles close to $\theta = 90°$, as seen in Fig. 2j,k.

## Discussion

In conclusion, we have analysed the effect of multiple scattering on the particle distributions resulting from spin-polarized $^{40}K$ collisions in an energy domain where every binary scattering event is exclusively p-wave. Contrary to the notion that fermionic clouds of $^{40}K$ would not have sufficient densities or inter-atomic interactions to readily observe scattering halos[27], we not only directly and clearly image such halos but also observe 90° scattering as a clear indication of multiple scattering. Our data, along with a numerical simulation of the experiment, enable us to classify the collisions into three regions based on the prevalence and effect of multiple scattering on the angular and radial distributions (Supplementary Note 3 discusses the radial aspect). Halos with a $\theta = 90°$ component belong to a region of isotropically enhanced multiple particle scattering, whereas a region with axial enhancement constitutes a domain for multiple scattering void of this; thus, a $\theta = 90°$ component is a sufficient, but not necessary, condition for multiple scattering.

Even for the non-degenerate system studied here, the consequences of a simple, fundamental, microscopic rule—the Pauli exclusion principle—when combined with high atom densities, lead to substantial modification of the macroscopic scattering halo from the binary collision expectation. The p-wave Wigner threshold defines a low-energy suppression region above which multiple scattering drives isotropically and axially enhanced modes, which are generally both present. Each of the modes of Fig. 4b,c can, nevertheless, be favoured individually through the functional dependence of $\sigma(E)$. In our particular experimental realization using $^{40}K$, the isotropically enhanced region stands out because of the p-wave Wigner threshold reigning energetically just below it, whereas the axially enhanced region singles out due to a shape resonance. It should be stressed, however, that axially favoured multiple scattering via the process of Fig. 4c is neither particular to $^{40}K$ nor to the p-wave nature of the scattering resonance but should happen for a broad class of elastic scattering resonance features in the collision of atomic clouds. Indeed, such resonances—a paradigm of scattering physics[2,10,28,29]—are known to exist in many systems in the realm of cold and ultracold collisions[30–32]. Moreover, the isotropically favoured mode we observe is a generic p-wave threshold phenomenon, which any system of indistinguishable fermions (with sufficiently short-range interactions[2]) should display.

In the future, multiple scattering in two-component collisions, such as between clouds in distinguishable spin states, could transform the bipartite entanglement generated by binary collisions[33] into multipartite entanglement. For this purpose, the insights gained in the present work may indeed prove valuable. On a more immediate note, we point out that experiments colliding dense atomic clouds to infer scattering phase shifts from the angular scattering distributions should ideally take the significant effects of multiple scattering into account. The analysis treatment via DSMC techniques employed in the work presented here may be a first step in this direction.

## Methods

**Experimental sequence.** We begin our experiment by collecting $^{87}Rb$ and $^{40}K$ atoms in a dual species magneto-optical trap and optically pumping the laser-cooled atoms to the $|F = 2, m_F = 2\rangle$ and $|F = 9/2, m_F = 9/2\rangle$ states, respectively. The atoms are transferred to a magnetic Ioffe–Pritchard trap where the K atoms are sympathetically cooled by forced evaporative cooling of the Rb atoms to $\approx 700\,nK$. The atoms are then transferred to an optical tweezer generated from a 1,064-nm Yb fibre laser. The laser power is divided between a static horizontal beam that provides vertical confinement and defines a collision axis, and a vertical beam that is used to move and accelerate the atoms along the horizontal beam[18]. The vertical beam passes through a two-axis acousto-optical deflector and by rapidly toggling between two driving frequencies we generate a pair of time-averaged traps. Atoms are separated by $5.80 \pm 0.06\,mm$ by adjusting the driving frequencies, the remaining Rb atoms are removed via a resonant light pulse and the K clouds are accelerated towards their midpoint. The trap is turned off when the clouds are separated by $80 \pm 2\,\mu m$ from the midpoint so that the clouds collide in free space. The trapping frequencies of the optical trap at the location where it is turned off are $(\omega_x, \omega_y, \omega_z) = 2\pi \times (250, 334, 384)\,s^{-1}$ with uncertainties of $\pm 2\pi \times 8\,s^{-1}$ and $z$ being the collision axis. We let the clouds separate by a minimum distance of $1.65 \pm 0.03\,mm$ before imaging. Collision energies are calibrated in separate measurements by measuring the distance travelled by the unscattered clouds in a set time; the systematic uncertainty in the energy is $\pm 8\,\mu K$ plus 2% of the value. All uncertainties are stated at the $1 - \sigma$ level. For details on imaging and analysis, see Supplementary Note 1.

**Direct-simulation Monte-Carlo.** The DSMC algorithm works by separating the motion of the atoms from their collisions, which is valid when the mean free path of an atom is much larger than other physical length scales such as the size of the atom clouds (a large Knudsen number)[26]. A single iteration of the DSMC algorithm first calculates the positions and velocities of all atoms by numerically integrating the equations of motion. One then assigns atoms to cells in a three-dimensional grid and calculates the probability of a collision between pairs of atoms using Bird's method[34]. If the probability is larger than a number $x$ randomly drawn from a uniform distribution $x \in [0, 1]$, then the collision succeeds and new relative velocities $\vec{v}_\pm = \pm v_{rel}/2(\cos\phi\,\sin\theta, \sin\phi\,\sin\theta, \cos\theta)$ in the centre-of-mass frame are generated randomly using a uniform distribution for $\phi \in [0, 2\pi]$ and the probability distribution $P(\theta) \propto d\sigma/d\Omega\,\sin\theta$ for $\theta$. Converting back to the laboratory frame yields the new velocities. Repeating these two steps, movement then collisions, effectively integrates the Boltzmann equation.

We initialize our simulation by generating $10^6$ test particles, which represent $8.2 \times 10^5$ physical particles and apportion them to one of two clouds with 10% more in the initial right-hand cloud as measured in experiment. These two clouds are separated by a mean distance of $80\,\mu m$ and have a mean relative velocity as determined by the desired seed energy. The individual test particles' positions and velocities are randomized about their clouds' mean values assuming Gaussian distributions given the experimentally measured temperatures and trapping frequencies. We then iterate the DSMC algorithm assuming no external forces up to the time-of-flight used for the experimental measurement for that particular seed energy. The final positions of the test particles are then binned and converted into a synthetic absorption image.

**Data availability.** The data that support the findings of this study are available from the corresponding author upon request.

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

## Acknowledgements

We thank Ina Kinski for manufacturing isotopically enriched $^{40}$K dispensers and Andrew Daley for comments on our manuscript. N.K. acknowledges the hospitality of the Johannes Gutenberg-Universität Mainz during the finalizing of the manuscript. This work was supported by the Marsden Fund of New Zealand (Contract Number UOO1121).

## Author contributions

K.O.R., A.B.D. and N.K. designed and constructed the optical collider. R.T. and K.O.R. performed experiments with support from A.B.D. R.T. analysed the data and calibrated the experimental apparatus. E.T. provided theoretical scattering matrices and cross-sections. R.T., A.C.J.W. and P.B.B. performed theoretical modelling of the data. R.T. and N.K. prepared the manuscript with input and comments from all authors. N.K. supervised the project.

## Additional information

**Competing financial interests:** The authors declare no competing financial interests.

