## [Peer Review File · Nature Communications]

Reviewer #1 (Remarks to the Author):

I reviewed this article for its previous submission. I find that the authors have appropriately changed the text to address the concerns of the referees.

With the additional details that the authors provide, it becomes clear that the angular resolution is not limited by the in-trap size but rather by the finite temperature of the colliding sample. In the $E/k=300$ uK case, the unscattered cloud subtends approximately 8 deg of the angular distribution. Taking this as a conservative estimate of the resolution limit, 8 deg may only cause a small offset (parameter b) on the order of $5/1000$ of the scattering distribution amplitude. Therefore the angular resolution is not a limiting factor.

I believe that Nature Communications is an appropriate journal for this study, and I recommend it for publication.

Reviewer #2 (Remarks to the Author):

I still do not consider the revised manuscript appropriate for publication in Nature Communications in the current form.

The authors study the multiple scatterings in their specific experimental conditions, but they do not derive any general features of the multiple scatterings from their experimental results. What they studied experimentally and reproduced theoretically is the multiple scatterings observed in the very specific experimental conditions, namely the collisions of potassium atoms with the shape resonance at 300 microkelvin. If the authors express their work as a clarification of the dynamics of the multiple scatterings, they should explicitly describe which aspect of the multiple scatterings is understood through the measurement. The classification of the collisional features in terms of energies is also very specific to the potassium atoms, and I do not think that the discussion given in the manuscript directly provides any new general insight into the dynamics of multiple scatterings.

I would also like to point out that the authors need to take the left-right asymmetry of the Fig. 2g more seriously. The "axial enhancement" is seen only at around 180 degree and not around 0 degree. If the left-right asymmetry of the data is due to the technical reason of the experiment as the authors comment, the enhancement at around 180 degree might just be due to this imperfection. As Referee 1 also mention in the Remark (3), only two markers near 180 degree support the consistency between the experiment and the theory, and the points near 0 degree seems to be consistent with both the simulation and a simple sinusoidal curve. Furthermore, I do not see consistency between the experiment and theory around 90 degree region, either. The data points do not match either with the simulation curve or the sinusoidal curve. It does not sound logical to claim that the data is consistent with theory by mentioning the relatively flat feature of the data near 90 degree region.

Reviewer #3 (Remarks to the Author):

The authors did quite some effort to discuss the points I raised in my previous report, and made changes accordingly in the manuscript. I am satisfied with these answers and changes. I believe

the current version of this paper is of high quality and represents highly original work. I recommend its publication in Nature Communications.

Authors' response to referee reports on the manuscript “Multiple scattering dynamics of fermions at an isolated p-wave resonance”

We are pleased to note that both Referee 1 and Referee 3 are satisfied by our first point-by-point response and the planned changes to our manuscript that we put forward. We are also happy to note that these referees both, unequivocally, recommend publication in Nature Communications. For example, Referee 1 notes that “*I believe the current version of this paper is of high quality and represents highly original work*”.

We have updated our manuscript according to aforementioned planned changes. In addition, slight reformatting has been carried out to meet the journal style. This includes a 150 word abstract followed by Introduction, Results, and Discussion.

Referee 2 takes the stand that “*I still do not consider the revised manuscript appropriate for publication in Nature Communications in the current form*” and raises two specific points — one regarding generality, and one regarding our interpretation of the data in Fig. 2g — which we will address below.

Point-by-Point response to remarks of Referee 2

- Referee #2, Remark (1) : *The authors study the multiple scatterings in their specific experimental conditions, but they do not derive any general features of the multiple scatterings from their experimental results. What they studied experimentally and reproduced theoretically is the multiple scatterings observed in the very specific experimental conditions, namely the collisions of potassium atoms with the shape resonance at 300 microkelvin. If the authors express their work as a clarification of the dynamics of the multiple scatterings, they should explicitly describe which aspect of the multiple scatterings is understood through the measurement. The classification of the collisional features in terms of energies is also very specific to the potassium atoms, and I do not think that the discussion given in the manuscript directly provides any new general insight into the dynamics of multiple scatterings.*

Response (2,1): Firstly, the multiple scattering which sets in at the Wigner threshold and adds an isotropic background is not specific to ^{40}K , but a generic p-wave threshold feature which, in particular, should apply to *any* system of indistinguishable fermions. Secondly, we find it noteworthy that for a collision of indistinguishable fermions the observation of 90° scattering is a sufficient, but not necessary condition. Notably, we observe that when a scattering resonance is present, multiple scattering *without* a 90° component can occur. The resonance at $300\ \mu\text{K}$ is indeed particular to ^{40}K and in response (3,4) of our original rebuttal letter we dealt in detail about the role it plays and what would happen in its absence. In particular it read

For pure p-wave scattering the two fundamental secondary collision processes of Fig. 4b and c (which lead to isotropic and axial enhancement of multiple scattering, respectively) are in principle always present. For an energy independent cross section both processes will happen with equal probability. The fortuitous occurrence of a shape resonance means that we can observe the effect of these processes individually. Had the resonance not been at our disposal, it would be possible to observe an isotropically added component on its own when approaching the suppression region from above, but the axially enhanced mode would not single out in the resonance free setting.

However, let us add to this that the enhancement of multiple scattering into axial modes is not facilitated by the resonance being p-wave, at $300\ \mu\text{K}$, or for ^{40}K . The mechanism of Fig 4c does not rely any of this. So thirdly, it is just the presence of *some* resonance which is important and scattering

resonances happens for many other systems than ^{40}K . To clarify this in our manuscript, we have expanded (highlighted in green below) the Discussion of our manuscript to read

In conclusion, we have analyzed the effect of multiple scattering on the particle distributions resulting from spin-polarized ^{40}K collisions in an energy domain where every binary scattering event is exclusively p-wave. **Contrary to the notion that fermionic clouds of ^{40}K would not have sufficient densities or inter-atomic interactions to readily observe scattering halos[27], we not only directly and clearly image such halos but also observe 90 degree scattering as a clear indication of multiple scattering.** Our data, along with a numerical simulation of the experiment, enable us to classify the collisions into three regions based on the prevalence and effect of multiple scattering on the angular and radial distributions (Supplementary Note 3 discusses the radial aspect). **Halos with a $\theta = 90^\circ$ component belong to a region of isotropically enhanced multiple particle scattering, whereas a region with axial enhancement constitutes a domain for multiple scattering void of this; thus a $\theta = 90^\circ$ component is a sufficient, but not necessary, condition for multiple scattering.**

Even for the non-degenerate system studied here, the consequences of a simple, fundamental, microscopic rule – the Pauli exclusion principle – when combined with high atom densities, lead to substantial modification of the macroscopic scattering halo from the binary collision expectation. The p-wave Wigner threshold defines a low-energy suppression region above which multiple scattering drives isotropically and axially enhanced modes, which are generally both present. **Each of the modes of Fig. 4b and Fig. 4c can, nevertheless, be favored individually through the functional dependence of $\sigma(E)$.** In our particular experimental realization using ^{40}K , the isotropically enhanced region stands out because of the p-wave Wigner threshold reigning energetically just below it, while the axially enhanced region singles out due to a shape resonance. It should be stressed, however, that axially favoured multiple scattering via the process of Fig. 4c is neither particular to ^{40}K nor to the p-wave nature of the scattering resonance but should happen for a broad class of elastic scattering resonance features in the collision of atomic clouds. Indeed, such resonances — a paradigm of scattering physics[3,26,28,29] — are known to exist in many systems in the realm of cold and ultracold collisions[30-32]. Moreover, the isotropically favoured mode we observe, is a generic p-wave threshold phenomenon which any system of indistinguishable fermions (with sufficiently short-range interactions[3]) should display.

In the future, multiple scattering in two-component collisions, such as between clouds in distinguishable spin states, could transform the bipartite entanglement generated by binary collisions[33] into multipartite entanglement. For this purpose the insights gained in the present work may, indeed, prove valuable. On a more immediate note, we point out that experiments colliding dense atomic clouds to infer scattering phase shifts from the angular scattering distributions should, ideally, take the significant effects of multiple scattering into account. The analysis treatment via DSMC techniques employed in the work presented here may be a first step in this direction.

• Referee #2, Remark (2) : *I would also like to point out that the authors need to take the left-right asymmetry of the Fig. 2g more seriously. The "axial enhancement" is seen only at around 180 degree and not around 0 degree. If the left-right asymmetry of the data is due to the technical reason of the experiment as the authors comment, the enhancement at around 180 degree might just be due to this imperfection. As Referee 1 also mention in the Remark (3), only two markers near 180 degree support the consistency between the experiment and the theory, and the points near 0 degree seems to be consistent*

Figure R1: Comparison between data, fits, and DSMC simulation. **a** Seed energy of $E/k = 920 \mu\text{K}$. **b** Seed energy of $E/k = 1380 \mu\text{K}$. Blue circles are experimental data, grey curves are sinusoidal fits, and red dashed lines are from the DSMC model. **c-d** Residuals normalized to experimental error for **a-b**, respectively. Grey circles are the residuals for the data and the sinusoidal fit, and red triangles are for the data and the DSMC model.

with both the simulation and a simple sinusoidal curve. Furthermore, I do not see consistency between the experiment and theory around 90 degree region, either. The data points do not match either with the simulation curve or the sinusoidal curve. It does not sound logical to claim that the data is consistent with theory by mentioning the relatively flat feature of the data near 90 degree region.

Response (2,2): The statement about axial enhancement is founded on observations at seven or so different energies throughout Region III and not only the original Fig. 2d ($E/k=1660\mu\text{K}$). To provide a more compelling argument, we have added an additional example from Region III to Fig. 2 ($E/k=1380\mu\text{K}$). Because this added example is for a lower collision energy, where the loss of atoms from the trap during acceleration is not as pronounced, the scattering halo is more symmetric. In particular, the comparison plot between data, fit, and simulation for a seed energy of $1380 \mu\text{K}$ shows how the sinusoidal fit does not provide as good of a description of the data as the DSMC model. We include in this response enlarged versions of the comparison plots for both $1380 \mu\text{K}$ and a lower energy of $920 \mu\text{K}$ as Fig. R1. We calculate the χ^2 value for each plot and each model type. For $920 \mu\text{K}$ we have $\chi_{\text{fit}}^2 = 73.5$ and $\chi_{\text{sim}}^2 = 15.6$ for 20 data points, and for $1380 \mu\text{K}$ we have $\chi_{\text{fit}}^2 = 108.1$ and $\chi_{\text{sim}}^2 = 22.8$ for 20 data points. From both a qualitative and quantitative perspective, the DSMC model is a better description of the experimental data than the sinusoidal fit.

We have referred to the energy range above $800 \mu\text{K}$ as the “axial enhancement” region based on both the larger-than-expected values of $dN_{\text{sc}}/d\Omega$ close to 0° and 180° , and the widening of the trough at 90° where the scattering distribution appears to be “pushed out” towards the collision axis. Referee 2 appears to be of the opinion that this does not constitute sufficient evidence to claim axial enhancement. Using smaller angular bins, and hence sacrificing some signal-to-noise, we can extend the analysis range without including the dense unscattered clouds such that the axial enhancement becomes more apparent in comparison to the suppression and isotropic enhancement regions, as seen in Fig. R2. Furthermore, since we keep track of collisions in the simulation we can create synthetic absorption images which do not contain the unscattered fraction, and using these images we can extend the analysis range nearly to the collision axis. We cannot extend it all the way to the collision axis

Figure R2: Scattering distributions with finer angular bins. **a-c** Experimentally measured scattering distributions (blue circles), sinusoidal fits (grey solid lines), and DSMC model (red dashed line) binned into 96 angular bins for seed energies E/k of 180, 300, and 1380 μK , respectively. The angular ranges for the experimental data are determined by the unscattered clouds' angular extent, whereas the angular range for the DSMC model is to within 5° of the collision axis.

because the BASEX method for Abel inversion creates significant noise along the $r = 0$ line. As can be seen in Fig. R2c, at 1380 μK , both the experimental data and the simulation show enhancement of scattering along the axial direction in contrast to Figs. R2a and b. We also note that $\chi_{\text{fit}}^2 = 190.6$ and $\chi_{\text{sim}}^2 = 49.9$ with 42 data points for Fig. R2c, demonstrating quantitatively that the DSMC model is a better description of the experimental data than the naive p-wave model, just as we have claimed in our manuscript. To alleviate confusion for future readers, we have reduced the size of the angular bins used for the analysis of Figure 2, so that the axial enhancement is more clearly seen.

We would like to stress that the asymmetry in the number of particles scattered left and right along the axis for the high energy case ($E/k = 1660 \mu\text{K}$) is *not* in contradiction with the mechanism (Fig.4c) put forward in our paper. In contrast, if the left (right) mother cloud (from which the once scattered particles predominantly scatters off a second time) is larger than the right (left), the left (right) axial mode is expected to dominate over the right (left) axial mode. As currently stated in our paper:

This effect is caused by the diffraction efficiency and mode structure of the acousto-optic deflector, which generates the time-averaged traps, not being entirely symmetric about the collision point and becomes pronounced when accelerating to high energies.

In addition to adding an example from region III to Fig. 2, we also added an example to region I, namely the collision energy $E/k = 180 \mu\text{K}$, which already received mentioning in our manuscript as being at the upper bound of the isotropically enhanced region.

Reviewer #2 (Remarks to the Author):

I appreciate the authors to revise the manuscript and incorporate the changes into the manuscript, which has made me easier to follow the logic of the discussion after the first revise. Now I am happy to recommend the paper for publication in Nature Communications.

As for the generality of their work, I understand the generality of their observation in the low collision energy regime, but I was not sure about the generality of the feature in the high collision energy regime. Now I understand by the sentence "... while the axially enhanced mode should occur in any colliding particle system with an elastic scattering resonance." This sentence makes it clear that the feature is general for the particles with a scattering resonance, which might have been implied in the previous manuscript but I did not get previously. I also like the way the authors revised the conclusion.

About the data of the angular distribution shown in Fig.2, I highly evaluate the way the authors revised the Fig.2. I do not think that adding the data at 1380 micro-K is not something superfluous, and I am now more convinced about the axial enhancement feature from two data in Fig.2j and 2k, particularly about the reproducibility and evolution of the scattering angular distributions. I appreciate the authors' effort to estimate the residuals and show the Figure R1 in the response. I am now fully convinced that their simulation matches with the data better than the simple cosine function.